# Kinetics of Low-Grade Scheelite Leaching with a Mixture of Sodium Phosphate and Sodium Fluoride

Liang Yang [1] , Chaoyang Li [1], Caifang Cao [1], Xiang Xue [2], Dandan Gong [1] and Linsheng Wan [1],*

[1] Faculty of Materials Metallurgy and Chemistry, Jiangxi University of Science & Technology, Ganzhou 341000, China

[2] Zijin Mining and Metallurgy Research Institute, Xiamen 361001, China

* Correspondence: 9119760008@jxust.edu.cn

**Abstract:** The current technology of leaching low-grade scheelite with sodium hydroxide or sodium carbonate has the disadvantages of large leaching reagent dosage and low leaching efficiency of tungsten. In order to extract scheelite efficiently, the kinetics of low-grade scheelite leaching with a mixture of sodium phosphate and sodium fluoride was investigated. In this study, the effects of temperature, phosphate concentration, and fluoride ion concentration on the leaching rate of tungsten were investigated. Our results showed that the leaching rate of tungsten was greatly influenced by the temperature and less affected by the concentration of phosphate and fluorine ions. The leaching process was controlled by a chemical reaction with an apparent activation energy value of $51 \pm 0.2$ kJ/mol. The apparent reaction orders with respect to phosphate and fluorine ions were 0.49 and 0.11, respectively. The reaction product calcium fluorophosphate was a loose, rod-like crystal, which would not tightly wrap on the surface of scheelite to prevent the diffusion process. The leaching kinetics of low-grade scheelite was in accordance with the shrinking core model, and the corresponding kinetic equation was also established.

**Keywords:** scheelite; sodium phosphate; calcium fluoride; calcium fluorophosphate; leaching kinetics





## 1. Introduction

Tungsten, as an important strategic metal, has been widely used in agricultural production, industry, and national defense and plays an indispensable, important role in the field of human economy and production because of its excellent physical, mechanical, and chemical properties [1,2]. The main economic minerals of tungsten are wolframite and scheelite. The reserves of scheelite are about two-thirds of the total reserves of tungsten ore in the world, and the grade of primary scheelite is most below 0.4%. In addition, it has been found that most of the tungsten ores discovered in recent years are low-grade scheelite through exploration [3–6]. With the gradual depletion of wolframite and high-grade scheelite concentrates, low-grade scheelite has gradually become the main raw material of the tungsten industry [7–10].

At present, globally speaking, scheelite is mainly leached with sodium hydroxide or sodium carbonate. In China, the process of leaching scheelite with sodium hydroxide is widely adopted [11–13]. However, the reaction equilibrium constant of leaching scheelite with sodium hydroxide is very small (25 °C, K = $2.5 \times 10^{-4}$), so it is necessary to increase the concentration of sodium hydroxide to promote scheelite decomposition. For scheelite concentrates (WO$_3$ > 65 wt%), the dosage of NaOH is generally 2.5~2.8 times the theoretical amount, while for low-grade scheelite (WO$_3$ < 40 wt%), the leaching efficiency is lower than 95% even if the dosage of sodium hydroxide is 4 times the theoretical amount [14]. In Western countries, sodium carbonate is widely used to digest scheelite. The reaction equilibrium constant of leaching scheelite with sodium carbonate is also small ($4.26 \times 10^{-1}$ at 25 °C). The leaching efficiency of low-grade scheelite reaches 98% only when the dosage

of sodium carbonate is as high as 5 times the theoretical amount [15–17]. Therefore, the current sodium hydroxide or sodium carbonate leaching process is not suitable for treating low-grade scheelite, due to the disadvantages of the large dosage of leaching reagent and low leaching efficiency. In recent years, researchers have proposed some new methods for treating low-grade scheelite. Scheelite was leached with a mixture of sulfuric acid and phosphoric acid based on the high solubility of phosphotungstic acid and transferred into the leaching solution in the form of phosphotungstic heteropoly acid. The results of the leaching kinetics of scheelite with the mixture of phosphoric acid and sulfuric acid showed that the apparent activation energy of the reaction was 63.8 kJ/mol, and scheelite leaching was controlled by a chemical reaction [18]. Scheelite leaching with a mixture of hydrochloric acid and hydrogen peroxide has also been proposed. Tungsten entered the leachate as a soluble peroxotungstic acid $(WO(O_2)_2(H_2O)_2)$. The leaching rate of scheelite was greatly affected by temperature and less affected by hydrochloric acid concentration [19].

Sodium phosphate is an effective decomposition reagent for scheelite. Hydroxyapatite (HAP) with a small solubility product is generated by leaching scheelite with sodium phosphate in an alkaline solution. The equilibrium constant of the reaction is $1.38 \times 10^{14}$ at 25 °C, indicating that the efficient decomposition of scheelite can be theoretically achieved under the conditions of low temperature and low sodium phosphate concentration. However, for the scheelite concentrate $(WO_3, 78wt\%)$, the leaching efficiency of tungsten reached 97% in the actual test when the dosage of NaOH was 1. 6 times the theoretical amount, and the dosage of $Na_3PO_4$ was 1.8 times the theoretical amount at 270 °C [20]. The experimental results did not agree with the thermodynamic analysis results of leaching scheelite with sodium phosphate. In order to determine the cause of this phenomenon, the kinetics of leaching scheelite with sodium phosphate was studied. It was found that the reaction product $Ca_5(PO_4)_3OH$ was very compact and wrapped on the surface of scheelite, which led to the internal diffusion process as the controlling step of the scheelite leaching reaction. Therefore, it was necessary to strengthen the internal diffusion process by increasing the sodium phosphate concentration and the temperature, so as to obtain a satisfactory leaching efficiency of scheelite. For low-grade scheelite, the leaching efficiency decreased significantly under the same conditions as those of leaching the scheelite concentrate with sodium phosphate in an alkaline solution. Therefore, a large amount of sodium phosphate and a high temperature are also necessary when leaching low-grade scheelite with sodium phosphate.

In view of the disadvantages of the large dosage of leaching reagent and low leaching efficiency existing in the current processes for leaching low-grade scheelite with sodium carbonate or sodium hydroxide or sodium phosphate, it is of great significance to propose a new method for the effective leaching of low-grade scheelite. Based on the compound solubility product and chemical reaction equilibrium theory, a new process for leaching low-grade scheelite with a mixture of sodium phosphate and calcium fluoride is presented. In the leaching process, the reaction product calcium fluorophosphate with a very small solubility product is generated (shown as Equation (1)), and low-grade scheelite can be effectively decomposed under the conditions of low leaching reagent dosage and low temperature.

$$9CaWO_4(s) + 6Na_3PO_4(aq) + CaF_2(s) = 9Na_2WO_4(aq) + 2Ca_5(PO_4)_3F(s) \tag{1}$$

The thermodynamic analysis results showed that the equilibrium constant of the reaction was as high as $7.71 \times 10^{36}$, which implied that low-grade scheelite can theoretically be completely decomposed. Our team studied the leaching of low-grade scheelite and investigated the effect of process parameters on the leaching efficiency of scheelite [21]. Our experimental results showed that the leaching efficiency of scheelite was 98.6% when the sodium phosphate stoichiometric ratio was 1.6, the calcium fluoride stoichiometric ratio was 1.0, and the temperature was 160 °C. The extraction of tungsten from a scheelite concentrate with phosphate and fluoride has been studied [22]. However, unlike scheelite concentrates, low-grade scheelite contains a large amount of calcite and other associated

minerals, which will also react with the leaching agent, thus interfering with the leaching of scheelite. In the leaching process of low-grade scheelite, the reaction product calcium fluorophosphate was an insoluble solid, which might be wrapped on the surface of the scheelite and hinder the mass transfer process. Therefore, it is necessary to study the kinetics of leaching low-grade scheelite, to determine the controlling step of the leaching process and to provide theoretical guidance for the efficient leaching of low-grade scheelite.

## 2. Materials and Methods

### 2.1. Materials and Reagents

The low-grade scheelite used in this study was provided by Ganzhou Haichuang Tungsten Industry Co., Ltd (Ganzhou, China). The ore was ground by a vibrating mill (Hengzhong brand, XZM-100) and used for the leaching experiments. The main chemical composition of low-grade scheelite is shown in Table 1. The X-ray diffraction analysis (XRD) results in Figure 1 show that the ore is mainly composed of scheelite ($CaWO_4$) and calcite ($CaCO_3$).

**Table 1.** Main components of low-grade scheelite.

| Element | WO$_3$ | Sn | Mo | S | Ca | Fe | Mn | P | As | SiO$_2$ |
|---|---|---|---|---|---|---|---|---|---|---|
| Content wt% | 37.52 | 0.69 | 0.14 | 1.13 | 12.26 | 1.81 | 0.34 | 0.13 | 0.32 | 2.31 |
| Relative error % | 0.68 | 1.23 | 1.41 | 0.81 | 0.42 | 1.72 | 1.43 | 1.52 | 2.11 | 0.45 |

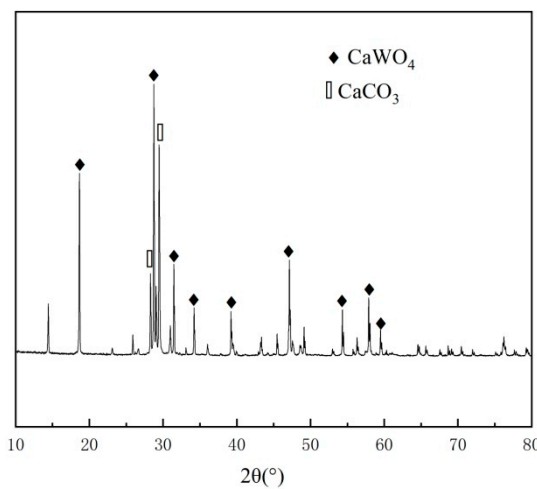

**Figure 1.** XRD patterns of low-grade scheelite.

The sodium phosphate, sodium hydroxide, and sodium fluoride used in this study were all analytical-grade reagents produced by Xilong Science Co., Ltd (Chengdu, China). All the aqueous solutions used in the study were prepared with deionized water.

### 2.2. Experimental Procedure

During the leaching process of low-grade scheelite, calcium fluoride first dissolved and released free fluoride ions, which reacted with phosphate ions and scheelite to form calcium fluorophosphate. To determine the apparent reaction order of these fluoride ions, it is necessary to maintain a fixed fluoride ion concentration during the leaching process. Since calcium fluoride is insoluble in water, soluble sodium fluoride was added in place of calcium fluoride to maintain a fixed fluoride ion concentration. During the process of leaching low-grade scheelite with the mixture of sodium phosphate and sodium fluoride, the addition of sodium hydroxide was necessary in order to maintain the alkalinity of the solution and ensured that phosphorus existed as orthophosphate. In this study, a certain amount of sodium hydroxide was added to adjust the pH value of the solution to around 12.

Briefly, a 400 mL solution with certain concentrations of sodium phosphate, sodium hydroxide, and sodium fluoride was added to an autoclave (YZPR-250 Intelligent microreactor). When the solution was heated to 95 °C, 4 g of low-grade scheelite was quickly added into the autoclave. The time started when the temperature in the autoclave reached a predetermined value. A large liquid–solid ratio (L/S = 100) was used in the leaching of low-grade scheelite to keep the leaching agent concentration approximately unchanged during the leaching process. Briefly, 3 mL of the slurry was removed from the autoclave at different times, and the $WO_3$ concentration of the leachate was determined to calculate the leaching efficiency of tungsten. The $WO_3$ concentration of each sample was determined twice. The thiocyanate colorimetric method was used to determine the concentration of tungsten using a spectrophotometer. Phosphorus concentration in the solution was determined using ICP-AES (Intrepid IIXSP). The fluoride ion concentration in the solution was determined by using a fluoride ion selective electrode. The leaching efficiency of tungsten X% is defined as:

$$X\% = (C_W \times V)/M_W \tag{2}$$

where $C_W$ represents the concentration of $WO_3$ in the leachate, g/L; V represents the volume of leachate, L; $M_W$ represents the mass of $WO_3$ in low-grade scheelite, g.

## 3. Results

### 3.1. Kinetic Model of Leaching Low-Grade Scheelite

The shrinking core model has been widely used to describe the leaching kinetics of minerals. In the leaching process of low-grade scheelite, the scheelite is gradually transformed into the reaction product calcium fluorophosphate, which surrounds the surface of scheelite particles. With the extension of the leaching time, the particle size of the scheelite is gradually reduced. Therefore, the shrinking core model was used to describe the leaching kinetics of low-grade scheelite in this study. According to several studies in the literature [23,24], the kinetic equation of the mineral leaching process can be expressed as Equation (3).

$$\frac{\delta\alpha}{3D_1} + \frac{r_0}{2D_s}[1 + \frac{2}{3}\alpha - (1-\alpha)^{\frac{2}{3}}] + \frac{1}{k_r}[1 - (1-\alpha)^{\frac{1}{3}}] = \frac{c_{A0}}{4\rho r_0}t \tag{3}$$

where $t$ represents the leaching time (s), $\alpha$ represents the leaching efficiency of tungsten (%), $\delta$ represents the thickness of the liquid film on a solid surface (m), $D_1$ represents the mass transfer coefficient of ion in the liquid film layer (m/s), $D_s$ represents the mass transfer coefficient of ion in the solid product layer (m/s), $k_r$ represents the chemical reaction rate constant, $\rho$ represents the molar density of the solid reactant (kmol/m$^3$), $c_{A0}$ represents the initial concentration of reactant $A$ (kmol/m$^3$), $r_0$ represents the radius of the low-grade scheelite particle (m).

When the leaching process is controlled by a chemical reaction, Equation (3) can be simplified to Equation (4).

$$1 - (1-\alpha)^{\frac{1}{3}} = \frac{k_r c_{A0}}{4\rho r_0}t \tag{4}$$

In comparison, when the leaching process is controlled by diffusion in the solid product layer, Equation (3) can be simplified to Equation (5).

$$1 - \frac{2}{3}\alpha - (1-\alpha)^{\frac{2}{3}} = \frac{D_s c_{A0}}{4\rho r_0^2}t \tag{5}$$

### 3.2. Influence of Temperature on the Leaching of Low-Grade Scheelite

The experiments were carried out with the temperature ranging from 100 °C to 130 °C for the −75/+58 µm mineral particle size with 0.7 mol/L $Na_3PO_4$ and 0.16 mol/L NaF in the solution, and the results are shown in Figure 2.

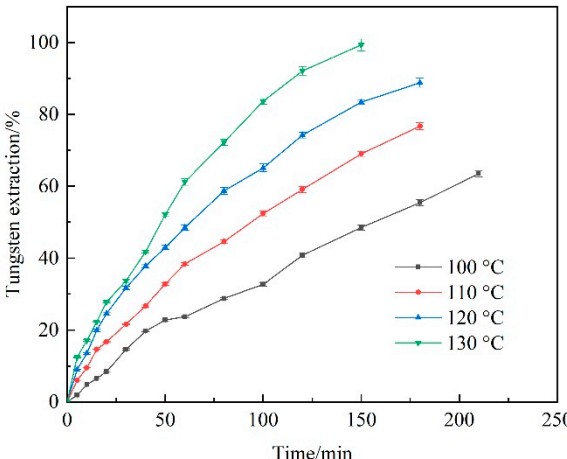

**Figure 2.** Effect of temperature on the leaching efficiency of tungsten.

It can be seen from Figure 2 that the temperature had a great influence on the leaching efficiency of scheelite. With the increase in temperature, the leaching efficiency of scheelite increased sharply. When the temperature was 130 °C, low-grade scheelite was completely decomposed within 150 min. Compared with leaching scheelite with sodium carbonate, the required temperature was obviously lower for leaching low-grade scheelite using the new process. Therefore, the decomposition of low-grade scheelite with the mixture of sodium phosphate and sodium fluoride can be promoted by increasing the temperature. In order to determine the leaching kinetic equation, the leaching efficiency of $WO_3$ at different temperatures was substituted into Equations (4) and (5), and the fitting correlation coefficient results are shown in Figures 3 and 4. It can be clearly observed that the chemical-reaction-controlled model fitted the experimental data of the leaching kinetics of low-grade scheelite better than the solid-product-diffusion-controlled model.

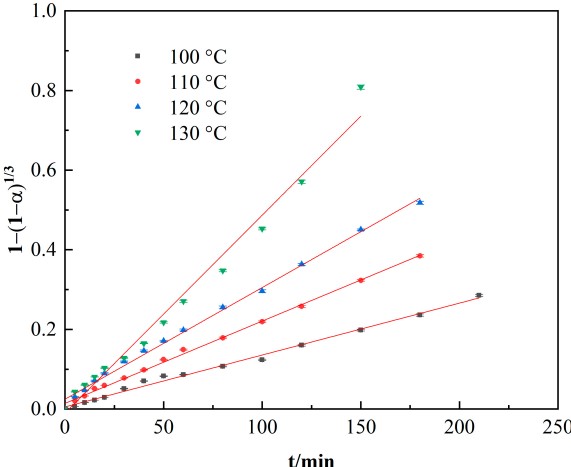

**Figure 3.** Relationship between $1 - (1 - \alpha)^{1/3}$ and time at different temperatures.

The apparent leaching rate constant k of tungsten at different temperatures can be determined as the slopes of the straight lines in Figure 3. Based on Equation (6), the Arrhenius plot of ln k vs. $T^{-1}$ is shown in Figure 5. The plot of k versus $1/T$ shows a favorable linear relationship. The apparent activation energy $E$ of the leaching reaction was calculated as 51.22 kJ/mol based on the slope of the line. The apparent activation energy $E$ was considered to be $51 \pm 0.2$ kJ/mol in consideration of the determination error. Hence, the leaching reaction of low-grade scheelite was considered to be controlled by a chemical reaction, since the apparent activation energy was greater than 42 kJ/mol [25].

$$k = A_0 \exp\left(\frac{-E}{RT}\right) \qquad (6)$$

where $A_0$ is the frequency factor, $E$ is the apparent activation energy (J/mol), $R$ is the universal gas constant (8.314 J/mol·K), and $T$ is the reaction temperature (K).

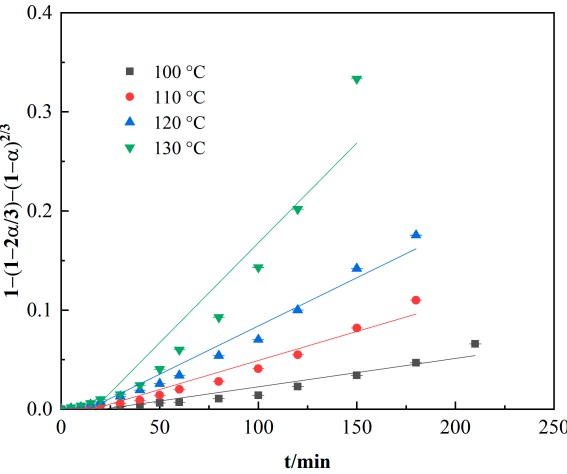

**Figure 4.** Relationship between $1 - (1 - 2\alpha/3) - (1 - \alpha)^{2/3}$ and time at different temperatures.

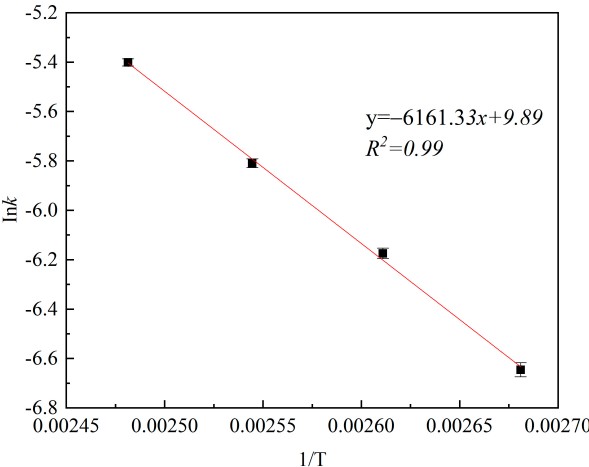

**Figure 5.** Relationship between ln$k$ and $1/T$.

### 3.3. Influence of Phosphate Concentration on Leaching of Low-Grade Scheelite

The experiments for investigating the influence of phosphate concentration on low-grade scheelite leaching were carried out with the phosphate concentration ranging from 0.5 mol/L to 0.9 mol/L at a temperature of 110 °C, a NaF concentration of 0.16 mol/L, and a mineral particle size of −75/+58 μm, and the results are shown in Figure 6.

It can be seen from Figure 6 that the increase in phosphate concentration promoted the leaching efficiency of low-grade scheelite. Compared with the temperature, the effect of sodium phosphate concentration on the leaching efficiency of tungsten was relatively smaller. The decomposition of low-grade scheelite is less dependent on the sodium phosphate concentration. This means that the efficient decomposition of scheelite can be achieved under the condition of a low sodium phosphate dosage. This has two advantages for low-grade scheelite leaching: On the one hand, a low sodium phosphate dosage reduces the cost, and on the other hand, the residual phosphorus concentration in the leaching solution is low, which is conducive to the subsequent phosphorus removal operation. In order to determine the apparent reaction order with respect to phosphate concentration, the leaching efficiency of tungsten under different phosphate concentrations is substituted

into Equation (4), which shows a good linear relationship between $1 - (1 - \alpha)^{1/3}$ and $t$ (Figure 7). In this study, the apparent reaction order for the sodium phosphate concentration was determined to be 0.49 (Figure 8), implying that the dependence of scheelite leaching on the $Na_3PO_4$ concentration is not strong.

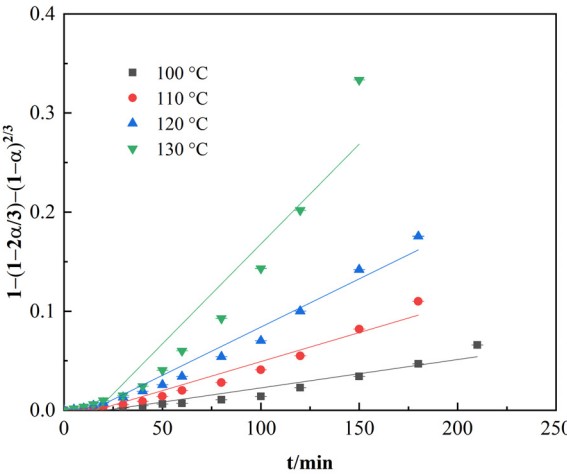

**Figure 6.** Effect of phosphate concentration on the leaching efficiency of tungsten.

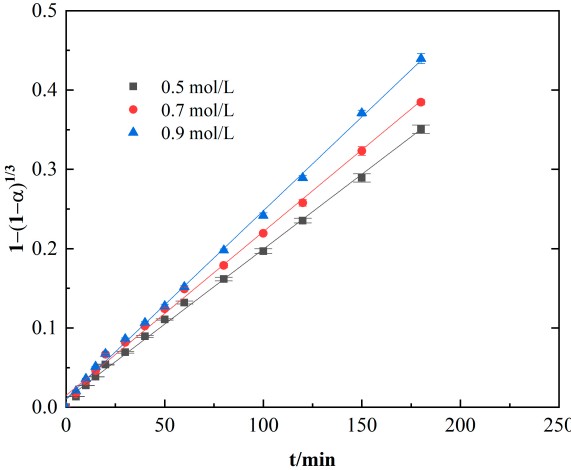

**Figure 7.** Relationship between $1 - (1 - \alpha)^{1/3}$ and $t$ at different phosphate concentrations.

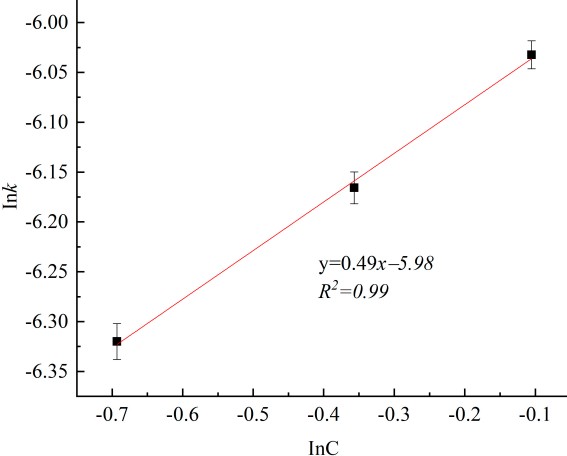

**Figure 8.** Plot of ln$k$ and ln$C_{Na3PO4}$.

### 3.4. Influence of Fluoride Ion Concentration on the Leaching of Low-Grade Scheelite

The effects of fluoride ion concentration on the leaching efficiency of scheelite were investigated with the fluoride ion concentration ranging from 0.05 to 0.5 mol/L at a temperature of 110 °C, a $Na_3PO_4$ concentration of 0.7 mol/L, and a mineral particle size of −75/+58 μm, and the results are shown in Figure 9.

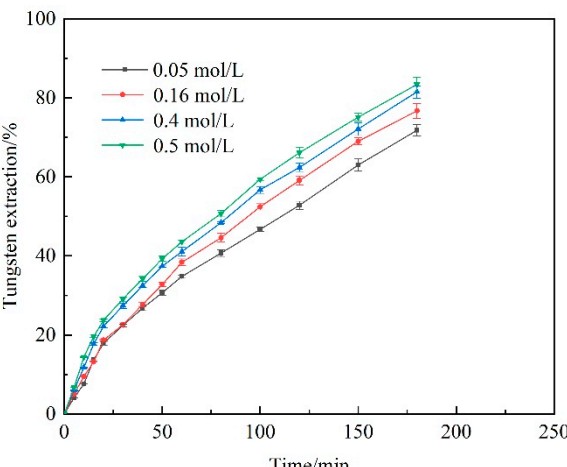

**Figure 9.** Effect of fluoride ion concentration on the leaching efficiency of tungsten.

It can be seen from Figure 9 that the leaching efficiency of scheelite increased with the increase in fluoride ion concentration. Compared with the temperature, the effect of fluoride ion concentration on the scheelite leaching efficiency was smaller. This means that scheelite can be efficiently decomposed at lower fluoride ion concentrations. Therefore, the addition of a small amount of calcium fluoride was enough for the efficient decomposition of scheelite. Figure 10 shows a favorable linear relationship between $1 − (1 − \alpha)^{1/3}$ and *t*. The apparent reaction order with respect to the fluoride ion concentration was determined to be 0.11 based on the slope of the fitting line in Figure 11, indicating a weak dependence on the fluoride concentration during the low-grade scheelite leaching process.

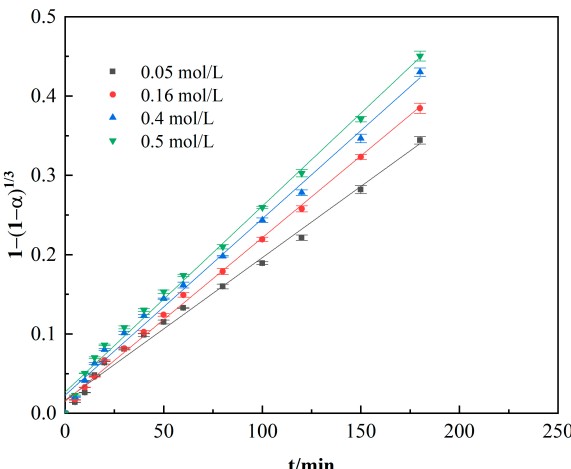

**Figure 10.** Relationship between $1 − (1 − \alpha)^{1/3}$ and *t* at different fluoride ion concentrations.

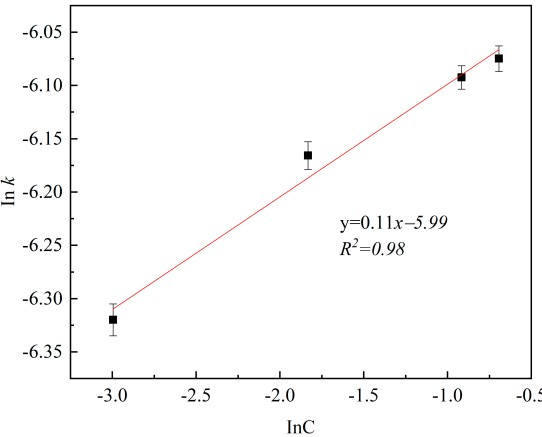

**Figure 11.** Plot of Ink and InC$_F$.

### 3.5. Influence of Mineral Particle Size on the Leaching of Low-Grade Scheelite

The effect of mineral particle size on the low-grade scheelite leaching was investigated with a sieved sample in the mixed solution, which contained Na$_3$PO$_4$ 0.7 mol/L and NaF 0.16 mol/L at 80 °C, and the results are shown in Figure 12.

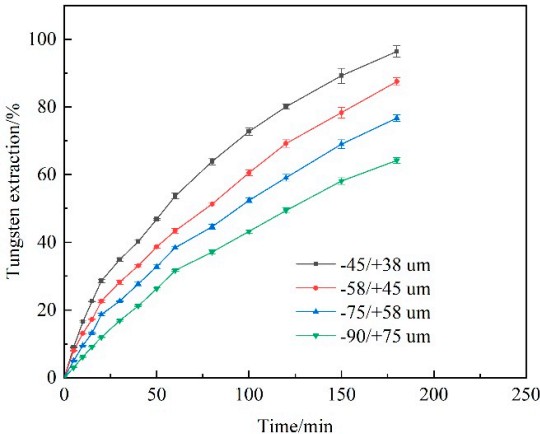

**Figure 12.** Effect of mineral particle size on the leaching efficiency of tungsten.

As shown in Figure 12, the particle size had a great influence on the tungsten leaching efficiency. With the decrease in the mineral particle size, the leaching efficiency of tungsten gradually increased. The experiment data were plotted according to Equation (4), and the results presented a favorable linear relationship, as shown in Figures 13 and 14. This further proved that the leaching of low-grade scheelite was controlled by a chemical reaction.

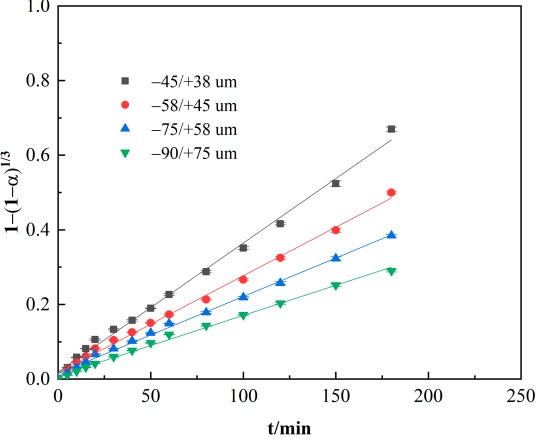

**Figure 13.** Relationship between $1 - (1 - \alpha)^{1/3}$ and $t$ at different mineral particle sizes.

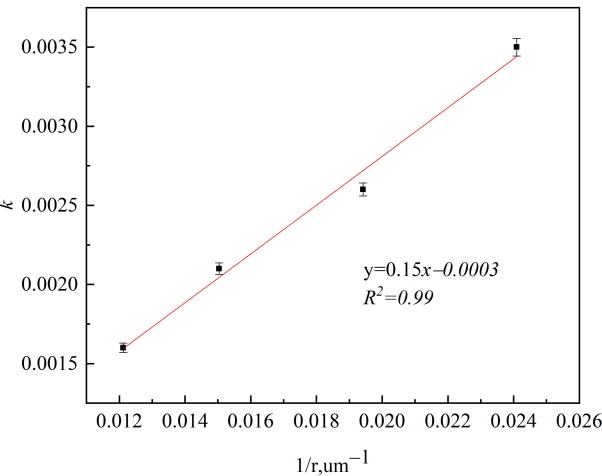

**Figure 14.** Plot of $k$ and $1/r$.

### 3.6. Establishment of Kinetic Equation

According to the above results of low-grade leaching kinetic, the kinetic equation can be expressed as:

$$1 - (1 - \alpha)^{\frac{1}{3}} = A_0 \cdot \exp\left(\frac{-51222}{RT}\right) \cdot C_{Na_3PO_4}^{0.49} \cdot C_F^{0.11} \cdot r^{-1} \cdot t \qquad (7)$$

According to Equation (7), all the experimental data were fitted, and the results showed a good linear relationship, with a correlation coefficient of 0.989. As shown in Figure 15, the slope of the fitted straight line is 0.032. Therefore, the kinetic equation of leaching low-grade scheelite can be expressed as:

$$1 - (1 - \alpha)^{\frac{1}{3}} = 3.2 \times 10^{-2} \cdot \exp\left(\frac{-51222}{RT}\right) \cdot C_{Na_3PO_4}^{0.49} \cdot C_F^{0.11} \cdot r^{-1} \cdot t \qquad (8)$$

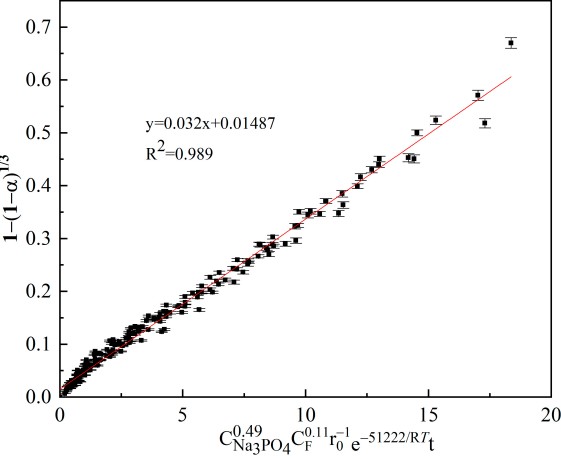

**Figure 15.** Relationship between $1 - (1 - \alpha)^{1/3}$ and $\exp\left(\frac{-51222}{RT}\right) \cdot C_{Na_3PO_4}^{0.49} \cdot C_F^{0.11} \cdot r_0^{-1} \cdot t$.

### 3.7. Analysis of Leaching Residue of Low-Grade Scheelite

In order to study the microstructure of the leaching residue and verify the controlling step of the low-grade scheelite leaching process, the leaching residue was analyzed using XRD (Figure 16) and SEM (Figure 17). It can be seen that the leaching residue was mainly composed of unreacted calcite ($CaCO_3$) and the reaction product calcium fluorophosphate ($Ca_5(PO_4)_3F$). It can be seen from Figure 17 that the calcium fluorophosphate product was a rod-like crystal and did not form a dense coating layer on the mineral surface, so the diffusion process of the leaching reaction was not hindered. The analysis results of the

leaching residue further confirmed the conclusion that the leaching process of low-grade scheelite was controlled by a chemical reaction.

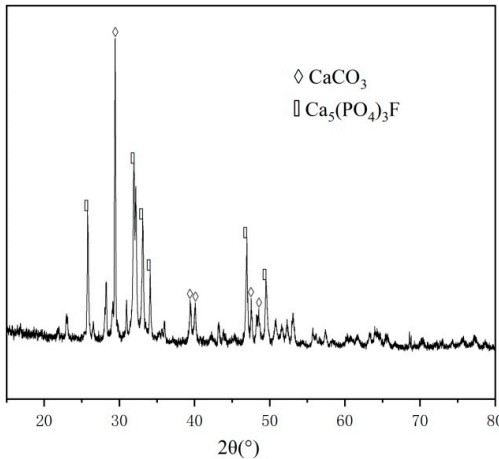

**Figure 16.** XRD pattern of the leaching residue.

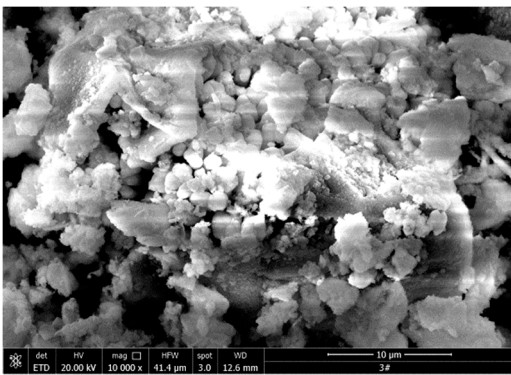

**Figure 17.** SEM of the leaching residue.

## 4. Conclusions

In this study, the leaching kinetics of low-grade scheelite with sodium phosphate and sodium fluoride was investigated. The experiment results showed that the leaching rate and leaching efficiency of scheelite were greatly affected by the temperature and less affected by phosphate and fluoride ion concentrations. Therefore, the leaching process of low-grade scheelite can be effectively strengthened by increasing the temperature. Since the concentrations of phosphate and fluoride ions have little effect on the leaching efficiency of scheelite, it is suggested to use lower concentrations of phosphate and fluoride ions to leach low-grade scheelite. The apparent activation energy E of the leaching low-grade scheelite reaction was $51 \pm 0.2$ kJ/mol, and the apparent reaction orders of sodium phosphate and fluoride ion were 0.49 and 0.11, respectively. The scanning electron microscope (SEM) and the XRD analysis results of the leaching residue showed that the reaction product was a loose, rod-like calcium fluorophosphate crystal, which would not hinder the mass diffusion process. These experimental data are consistent with the shrinking core model, indicating that the leaching process of low-grade scheelite was controlled by a chemical reaction.

**Author Contributions:** L.Y.: investigation, methodology, writing—original draft preparation; C.L.: investigation, validation, writing—review and editing; C.C.: supervision, validation; X.X.: formal analysis; D.G.: supervision, writing—review and editing; L.W.: methodology, data curation, project administration, resources. All authors have read and agreed to the published version of the manuscript.

**Funding:** This study was supported by the National Natural Science Foundation of China (No. 5196040227), the Jiangxi Province High-End Leading Talents Cultivation Project (No. 20204BCJ23001), the National Natural Science Foundation of Jiangxi (No. 20212BAB204027, No. 20212BAB214024), and Key Projects of Jiangxi Key R&D Plan (No. 20192ACB70017)).

**Institutional Review Board Statement:** Not applicable.

**Informed Consent Statement:** Not applicable.

**Data Availability Statement:** Not applicable.

**Conflicts of Interest:** The authors declare no conflict of interest.

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
