# Peer review of "Kinetics of Low-Grade Scheelite Leaching with a Mixture of Sodium Phosphate and Sodium Fluoride"

_metals, doi:10.3390/met12101759_

Round 1

Reviewer 1 Report

The authors clearly present a study of the influence of selected parameters (temperature, concentration of selected substances) on scheelite leaching. The results of the experiments and the kinetic analysis are clearly presented, but there is a complete lack of analysis of the errors of the experiments themselves, the methods used to determine the concentration of substances (or conversion) and the subsequent kinetic calculations. That is the reason to propose a minor revision.

Remarks to authors:

It is necessary to indicate the errors of determination for data in: Table 1, Figs. 2-15      and these errors should be reflected in the kinetic analysis parameter values. I really don't believe that the error of determining E is in the second decimal place, when it is usually units - so then surely the result is not E value 51.22 but 51 +/-... kJ/mol   Page 4, paragraph below equation 3: the symbols in the text do not correspond to the given equation and alpha is completely missing  

Page 5, line 172: there should be Fig. 2 not Fig. 4

Page 6, line 195: you mention an E value of 42 kJ/mol, where did this value come from?

Page 6, line 197: extra slash in units (J/mol/)

Fig. 15, x-axis label: why is there another symbol in the subscript for temperature when it is not in the equation?

Author Response

  1. The authors clearly present a study of the influence of selected parameters (temperature, concentration of selected substances) on scheelite leaching. The results of the experiments and the kinetic analysis are clearly presented, but there is a complete lack of analysis of the errors of the experiments themselves, the methods used to determine the concentration of substances (or conversion) and the subsequent kinetic calculations. That is the reason to propose a minor revision.

Remarks to authors:

Point 1: It is necessary to indicate the errors of determination for data in: Table 1, Figs. 2-15    …  and these errors should be reflected in the kinetic analysis parameter values. I really don't believe that the error of determining E is in the second decimal place, when it is usually units - so then surely the result is not E value 51.22 but 51 +/-... kJ/mol  Page 4, paragraph below equation 3: the symbols in the text do not correspond to the given equation and alpha is completely missing 

Response 1: Errors in the determination of data in Table 1 and Figs.2-15 have been added and are reflected in the figure as error bars. Allow for determination error, the apparent activation energy E has been revised as 51±0.2 kJ/mol. The symbols in the text have been revised correspond to the given equation 3 and alpha has been added.

Point 2: Page 5, line 172: there should be Fig. 2 not Fig. 4

Response 2: The Fig.4 has been revised as Fig.2 in page 5, line 172.

Point 3: Page 6, line 195: you mention an E value of 42 kJ/mol, where did this value come from?

Response 3: According to reference, it is generally considered that the apparent activation energy E of a reaction controlled by chemical reaction is greater than 42 kJ/mol. The reference has been cited in the manuscript.

Point 4: Page 6, line 197: extra slash in units (J/mol/)

Response 4: The extra slash in units (J/mol/) has been deleted.

Point 5: Fig. 15, x-axis label: why is there another symbol in the subscript for temperature when it is not in the equation?

Response 5: In the x-axis label of Fig. 15, T represents temperature, t represents reaction time.

Reviewer 2 Report

The manuscript “Kinetics of low-grade scheelite leaching with a mixture of sodium phosphate, sodium hydroxide and calcium fluoride” reported the effects of temperature, phosphate concentration and fluoride ion concentration on the leaching rate of tungsten for low-grade scheelite. This manuscript looks so similar to the papers published recently by the same authors [Process Safety and Environmental Protection, 2022, 159, 708-715; Minerals Engineering, 2022, 177, 107372], which were not even mentioned in the introduction section for the literature review. Especially, the same experimental data and even the same figure appeared in the previous papers and this manuscript, such as Fig. 17, which also the Fig 16a in Minerals Engineering, 2022, 177, 107372. Therefore, this manuscript is not recommended for publication.

Author Response

Point 1: The manuscript “Kinetics of low-grade scheelite leaching with a mixture of sodium phosphate, sodium hydroxide and calcium fluoride” reported the effects of temperature, phosphate concentration and fluoride ion concentration on the leaching rate of tungsten for low-grade scheelite. This manuscript looks so similar to the papers published recently by the same authors [Process Safety and Environmental Protection, 2022, 159, 708-715; Minerals Engineering, 2022, 177, 107372], which were not even mentioned in the introduction section for the literature review. Especially, the same experimental data and even the same figure appeared in the previous papers and this manuscript, such as Fig. 17, which also the Fig 16a in Minerals Engineering, 2022, 177, 107372. Therefore, this manuscript is not recommended for publication.

Response 1:  In this paper (Process Safety and Environmental Protection, 2022, 159, 708-715), the extraction of tungsten from scheelite concentrate with phosphate and fluoride has been studied. However, unlike scheelite concentrate, low-grade scheelite contains a large number of calcite and other associated minerals, which will also react with the leaching agent, thus interfering with the leaching of tungsten. Thus, the leaching processes of scheelite concentrate and low-grade scheelite are different. In the previous period, our team studied the leaching of low-grade scheelite, and investigated the effect of process parameters on the leaching efficiency of scheelite. The relative research results are published in the Journal of Mineral Engineering (Minerals Engineering, 2022, 177, 107372). However, this study does not include the content of leaching kinetics of low-grade scheelite. Therefore, this study is different from the current one (to be published in the journal metals).

Reviewer 3 Report

You can find my corrections and suggestions below:

1.      In lines 85-86, as you mentioned, there are three leaching reagents (sodium phosphate, sodium hydroxide and calcium fluoride) but in the equation (line 90) there are two reagents. Please explain it in the text.

2.      In line 105, please add brand and model for vibrating mill.

3.      Table 1 in line 112 can be centered like the rest of the manuscript.

4.      There is a little confusion in lines 120-121. You stated that calcium fluoride was used in the experiment in the whole manuscript but there you said you used sodium fluoride in place of calcium fluoride. However, you stated in the abstract that you used sodium phosphate, sodium hydroxide and calcium fluoride.

5.      In lines 123-124, you mentioned that you used sodium hydroxide as controlling agent but in the previous texts (abstract and end of the introduction) it sounds like you used it as leaching reagent. Please edit the text in more understandable way.

6.      In line 134, there is a term “regular time intervals”. Can you specify that?

7.      In lines 127-128, again you said you added sodium phosphate, sodium hydroxide and sodium fluoride into autoclave. How did you obtain calcium fluorophosphate (line 148)?

8.      The constant condition below the tables can be written in the text or you can add the term constant conditions before them (lines 170-207-233-252).

9.      In line 178, you again stated that sodium phosphate, sodium hydroxide and calcium fluoride are used in the mixture in despite of before.

10.  According to your results, since phosphate and fluoride ion concentration is not so effective, did you do experiments with the least ion concentration (Na3PO4: 0.5 mol/L ; NaF: 0.5 mol/L) at different temperatures? If you didn't, maybe you can add it in the conclusion as suggestion.

Author Response

Point 1: In lines 85-86, as you mentioned, there are three leaching reagents (sodium phosphate, sodium hydroxide and calcium fluoride) but in the equation (line 90) there are two reagents. Please explain it in the text.

Response 1: As shown in the equation (line 90), there two reagents (sodium phosphate, calcium fluoride)for low-grade scheelite. Sodium hydroxide is used as a controlling agent. It has been revised in the manuscript.

Point 2: In line 105, please add brand and model for vibrating mill.

Response 2: The brand and model for vibrating mill have been added.

Point 3: Table 1 in line 112 can be centered like the rest of the manuscript.

Response 3: Table 1 in line 112 has been centered.

Point 4:. There is a little confusion in lines 120-121. You stated that calcium fluoride was used in the experiment in the whole manuscript but there you said you used sodium fluoride in place of calcium fluoride. However, you stated in the abstract that you used sodium phosphate, sodium hydroxide and calcium fluoride.

Response 4: In the abstract and conclusion, the reagents “sodium phosphate, sodium hydroxide and calcium fluoride” have been revised as “sodium phosphate and sodium fluoride”. In the Experimental procedure part, we explained the reason of using sodium fluoride instead of calcium fluoride.

Point 5: In lines 123-124, you mentioned that you used sodium hydroxide as controlling agent but in the previous texts (abstract and end of the introduction) it sounds like you used it as leaching reagent. Please edit the text in more understandable way.

Response 5: We have been edited the manuscript in in more understandable way. The abstract, conclusion and end of the introduction have been revised. Sodium hydroxide was actually used as a controlling agent.

Point 6: In line 134, there is a term “regular time intervals”. Can you specify that?

Response 6: The term “regular time intervals” has been revised as “different time”.

Point 7: In lines 127-128, again you said you added sodium phosphate, sodium hydroxide and sodium fluoride into autoclave. How did you obtain calcium fluorophosphate (line 148)?

Response 7: After addition of sodium phosphate, sodium hydroxide and sodium fluoride into the autoclave, low-grade scheelite was subsequently added to the autoclave. These reagents reacted with scheelite to form calcium fluorophosphate precipitate. The ionic equation of this reaction can be expressed as follow: 9CaWO4 + 6PO43- + 2F- = 2Ca5(PO4)3F + 9WO42-

Point 8: The constant condition below the tables can be written in the text or you can add the term constant conditions before them (lines 170-207-233-252).

Response 8: The constant condition below the tables has been written in the text.

Point 9: In line 178, you again stated that sodium phosphate, sodium hydroxide and calcium fluoride are used in the mixture in despite of before.

Response 9: In the manuscript, it has been revised as sodium phosphate and sodium fluoride.

Point 10: According to your results, since phosphate and fluoride ion concentration is not so effective, did you do experiments with the least ion concentration (Na3PO4: 0.5 mol/L ; NaF: 0.5 mol/L) at different temperatures? If you didn't, maybe you can add it in the conclusion as suggestion.

Response 10: Thank you for your suggestion. The experiments with the lower ion concentration have been added in the conclusion as suggestion.

Round 2

Reviewer 2 Report

The authors have properly answered the questions from reviewers and made great improvements to the manuscript. It could be published as it is.

Reviewer 3 Report

All corrections and recommendations have been done. This manuscript can be published in this form.